# Trends, Epidemiology, and Management of Multi-Drug Resistant Gram-Negative Bacterial Infections in the Hospitalized Setting

**DOI:** 10.3390/antibiotics9040196

**Published:** 2020-04-20

**Authors:** Sabrina Morris, Elizabeth Cerceo

**Affiliations:** 1Cooper Medical School of Rowan University, Camden, NJ 08103, USA; morriss3@rowan.edu; 2Department of Hospitalist Medicine, Cooper University Hospital, Camden, NJ 08103, USA

**Keywords:** antibiotic resistance, antibiotic stewardship, Gram-negative bacteria

## Abstract

The increasing prevalence of antibiotic resistance is a threat to human health, particularly within vulnerable populations in the hospital and acute care settings. This leads to increasing healthcare costs, morbidity, and mortality. Bacteria rapidly evolve novel mechanisms of resistance and methods of antimicrobial evasion. *Escherichia coli*, *Pseudomonas aeruginosa*, *Klebsiella pneumoniae*, and *Acinetobacter baumannii* have all been identified as pathogens with particularly high rates of resistance to antibiotics, resulting in a reducing pool of available treatments for these organisms. Effectively combating this issue requires both preventative and reactive measures. Reducing the spread of resistant pathogens, as well as reducing the rate of evolution of resistance is complex. Such a task requires a more judicious use of antibiotics through a better understanding of infection epidemiology, resistance patterns, and guidelines for treatment. These goals can best be achieved through the implementation of antimicrobial stewardship programs and the development and introduction of new drugs capable of eradicating multi-drug resistant Gram-negative pathogens (MDR GNB). The purpose of this article is to review current trends in MDR Gram-negative bacterial infections in the hospitalized setting, as well as current guidelines for management. Finally, new and emerging antimicrobials, as well as future considerations for combating antibiotic resistance on a global scale are discussed.

## 1. Introduction

Antibiotic resistance is a growing problem around the world with an estimated 2.8 million antibiotic resistant infections occurring per year in the United States alone. These infections are associated with increased morbidity, mortality, and healthcare costs and lead to approximately 35,000 deaths each year. While Gram-positive bacteria such as methicillin resistant *Staphylococcus aureus* and *Clostridium difficile* were previously most concerning in the world of antibiotic resistance, currently, drug resistant Gram-negative bacteria are on the rise in the U.S. and around the globe. The Centers for Disease Control (CDC) developed a list of emerging antibiotic resistance threats first in 2013 and updated that list in 2019. The list is organized by level of urgency. “Urgent threats” are those that are imminent problems and require immediate action to prevent and treat. In 2013, the Gram-negative bacteria on this list included carbapenem resistant *Enterobacteriaceae* and cephalosporin resistant *Neisseria gonorrhoeae*. “Serious threats” are those that need monitoring as they have the ability to become imminent threats to public health. Those Gram-negative bacteria included multi-drug resistant (MDR) *Acinetobacter*, extended spectrum βlactamase producing *Enterobacteriaceae* (ESBL), MDR *Pseudomonas aeruginosa*, drug resistant *salmonella*, and *shigella* [1,2]. As of 2015, *E. coli*, *K. pneumoniae*, and *P. aeruginosa* accounted for 70% of Gram-negative hospital infections [3]. “Concerning bacteria” are deemed lowest priority on the list and were entirely Gram- positive bacteria in 2019. The latest report, released in November of 2019, is not encouraging. Carbapenem resistant *Acinetobacter baumannii* and *Enterobacteriaceae* spp. have moved to the top of the list with an estimated 281 million dollars in U.S. healthcare costs in 2017. Carbapenem resistant *Enterobacteriaceae* contributed roughly 130 million dollars alone to that total, and unfortunately, the patients most at risk for these infections are already hospitalized with indwelling devices or taking extended courses of antibiotics. ESBL *Escherichia coli*, *Klebsiella pneumoniae*, and MDR *Pseudomonas aeruginosa* maintained their status of “serious threats” in 2019 [2]. Antibiotic resistance collectively has increased U.S. healthcare costs in excess of 55 billion dollars annually when loss of productivity is included in the estimate. Total hospital costs are 29% higher (*p* < 0.0001) when treating patients with MDR pathogens given the longer median of stay and increased use of resources [3,4].

The root of the rise of antimicrobial resistance is multi-factorial. A key factor in emerging resistance is a lack of good antibiotic stewardship leading to overuse of antimicrobials, inappropriate empiric coverage, and delays in accurate diagnoses, as well as de-escalation of therapy. As time goes on, there are fewer and fewer antimicrobials available that are effective in treating these infections, and so, the problem further escalates. This issue is becoming especially problematic in hospitals and acute care settings. Among hospital acquired infections (HAI), drug resistant Gram-negative bacteria are becoming increasingly prevalent. *Klebsiella pneumoniae*, *Pseudomonas aeruginosa*, *Escherichia coli*, and *Acinetobacter baumannii* make up the majority of Gram-negative HAI in the U.S., and they are becoming exceedingly resistant to antibiotics. There is a variety of mechanisms by which these bacteria become resistant, which include, but are not limited to, producing β-lactamases and extended-spectrum β-lactamases, carbapenemases, and by increasing efflux activity [5]. These bacteria are often residents of acute care and long-term facilities and are responsible for pneumonias, catheter site blood stream infections, intraabdominal infections, and urinary tract infections (UTI). The patient population in these settings tends to be vulnerable to infection and often has multiple comorbidities, making them especially dangerous. A sampling of the hands of 125 healthcare workers in the intensive care unit (ICU), medicine floors, and surgical units in Greece found high rates of *Staphylococcus* spp. and *Acinetobacter baumannii* colonization, particularly among nurses, indicating that proper hand hygiene also plays a role in infection prevention [6]. The SENTRY trial recently identified these bacteria, among many others, to be significantly less susceptible to antimicrobials when isolated from the ICU versus other hospital units [7]. To further complicate the problem, therapies that are effective at targeting these organisms are dwindling at an alarming rate. There have been few developments in the last decade in antimicrobials, particularly for Gram-negative bacteria. Physicians have been forced to rely on older antibiotics such as colistin and fosfomycin, which have serious side effects and are now also seeing resistance [8]. Given the increase in morbidity, mortality, and healthcare burden that these organisms are responsible for, it is imperative to focus on infection prevention, adequate infection management, and the development of novel, targeted Gram-negative bacterial therapies. 

## 2. Risk Factors for Resistance 

Resistance is rising, particularly among patients in acute care settings such as intensive care units, which contributes to the extremely high mortality rates ranging between 26% and 80%. The World Health Organization determined that carbapenem resistance and ESBL producing infections were top priority in terms of the research and development of new drugs and innovative diagnostic techniques. In learning how to treat resistant pathogens properly, it is essential to identify patients most at risk for these infections in order to avoid delays in adequate treatment and the unnecessary use of strong empiric antibiotics [9]. ESBL producing pathogens are becoming increasingly common in both the hospitalized setting and the community. A retrospective case-controlled study in Australia evaluated the risk factors for the development of these bacterial infections [10]. They determined that the greatest predictors of ESBL production were length of stay (LOS) prior to infection (*p* < 0.0001), exposure to antibiotics within the last six weeks (*p* < 0.001), recent return from travel abroad (particularly in Asia) (*p* < 0.03), admission to the ICU (*p* < 0.001), and finally, residence in a long-term care facility (*p* > 0.001) [11]. These findings are consistent with similar literature. Research in a French ICU to determine predictive factors for the development of ESBL Gram-negative pathogens (GSB) infections during hospitalization additionally found male sex, age >75 years old, exposure to a third-generation cephalosporin or β-lactam in the prior three months, and colonization pressure of the unit were risk factors [12]. Finally, a one year, prospective surveillance study of MDR GNB blood stream infections also identified male sex, older age (>60), and co-morbidity (Charlson score) to be independent risk factors for resistance. They additionally determined that some pathogens were more likely to develop resistance than others such as *Enterobacteriaceae* spp., which were positively associated with multi-drug resistance. *K. pneumoniae* alone had an odds ratio (OR) of 4.59 for the development of resistance. Greater than 50% of *K. pneumoniae* isolates in the study were resistant to ampicillin-sulbactam, piperacillin-tazobactam, ceftriaxone, ceftazidime, cefepime, and ciprofloxacin, while 2% were also resistant to colistin [13]. Another study found that for each hospital-acquired *K. pneumoniae* infection, the risk of developing a drug resistant *K. pneumoniae* infection in subsequent hospitalizations increased 14% [14]. 

Identification of these risk factors can be crucial in the fight against MDR GNB. An appropriate level of clinical suspicion can ensure timely and adequate empiric therapy for patients at highest risk. Similarly, patients without identifiable risk factors can be safely spared the exposure to overly broad and powerful antibiotics and the subsequent toxicities. Including this type of screening into practice will ultimately lead to the more judicious use of antimicrobials, which is perhaps our greatest defense in the fight against resistance for the time being. 

## 3. Empiric Treatment Guidelines 

Antimicrobial therapy should always be tailored to the identified pathogen and subsequent susceptibility testing. Additionally, consideration should always be given to individual patient characteristics such as drug allergies and kidney and liver function. In reality, though, empiric therapy is often necessary to stave off life-threatening infections, especially in the vulnerable population of patients within the ICU and acute care facilities. Empiric treatment should be broad enough to cover pathogens most commonly associated with the particular infection at hand, while narrowed to the pathogens specific to the institution [5]. Overly broad coverage comes with the increased risk of resistance and exposure to the toxicities associated with therapeutic agents. Proper pathogen identification and susceptibility testing should be prompt, and therapy should be narrowed and de-escalated whenever possible. Special attention should be paid to the appropriate dosing and duration of therapy to ensure adequate coverage with minimal harmful exposure [5,15]. The following are guidelines for the diagnosis and empiric treatment of the most common hospital acquired infections. While specific pathogens and therapies are mentioned, it is always advisable to follow an antibiogram specific to the institution in order to best predict pathogens and resistance patterns. 

### 3.1. Hospital Acquired Pneumonia and Ventilator Associated Pneumonia 

Pneumonias are among the most common hospital acquired infections. Previously, all healthcare associated pneumonias were grouped under the term healthcare associated pneumonia (HCAP). This designation was based on the assumption that any pneumonia that developed following exposure to healthcare facilities (hospitalization within the last 90 days, residents of long-term care facilities, and those receiving chemotherapy or IV antibiotics) carried the same risk of resistance [16]. We now understand that while use of healthcare facilities does expose an individual to resistance, the individual risk factors of that patient, such as length of hospitalization, comorbidities, and use of invasive devices such as endotracheal tubes, are more indicative of the risk of resistance. Additionally, studies of the efficacy of the HCAP model showed no improvement in outcomes and in fact led to the unnecessary use of broad-spectrum antibiotics. As such, HCAP was divided into two similar, but distinct groups: hospital acquired pneumonia (HAP) and ventilator associated pneumonia (VAP). HAP is defined as a pneumonia acquired within 48 h of admission to the hospital, while VAP is a pneumonia that develops after 48–72 h of intubation [16,17]. The distinction is an acknowledgement of the variability among pathogens associated with each setting. Timing of infection can also be a clue in pathogen identification as infections acquired early on in admission are far more likely to be susceptible to antimicrobials than those acquired later. 

Diagnosis of HAP can be made based on clinical criteria and confirmed with imaging such as a chest radiograph. Treatment is typically empiric, and consideration of hospital specific pathogen and resistance patterns should be given. VAP, on the contrary, warrants microbiological data for pathogen identification and sensitivity data. The Infectious Diseases Society of America (IDSA) guidelines recommend non-invasive endotracheal aspirate sampling with semi-quantitative cultures. Empiric treatment should cover *Staphylococcus aureus*, *Pseudomonas aeruginosa*, and other Gram-negative bacteria such as *Klebsiella pneumoniae* and *Escherichia coli* while culture data are pending. Methicillin resistant *Staphylococcus aureus* (MRSA) coverage is not recommended unless the patient has known factors for resistance or there are high rates of MRSA infection within the unit or MRSA data within the unit are unknown. Methicillin-sensitive *Staphylococcus aureus* (MSSA) can be adequately covered with oxacillin, nafcillin, or cefazolin. Should MRSA coverage be needed, IDSA recommends vancomycin or linezolid. Monotherapy is appropriate for *Pseudomonas aeruginosa* when the patient is not in septic shock or at high risk of death. *Acinetobacter baumannii* is a bacterium of particular concern as resistance continues to grow worldwide. Treatment with imipenem or ampicillin-sulbactam is preferred if susceptible. Colistin should be reserved for infections resistant to the aforementioned therapy given its nephrotoxic profile. Use of tigecycline is not recommended as studies have shown decreased cure rates when compared with imipenem and higher mortality than colistin [5,17].

### 3.2. Blood Stream Infections

Bloodstream infections (BSI) are particularly offensive given their propensity to progress to severe sepsis and septic shock. As such, they are associated with high morbidity, increased healthcare costs, length of stay, and, of course, mortality. Prevention, early detection, and appropriate therapeutic strategies are therefore crucial in management. These infections can be categorized as primary or secondary. Primary is a confirmed bacteremia with no obvious source of infection such as skin and soft tissue or urinary tract infection (UTI). BSI are primarily healthcare associated infections with very few arising from the community. Indwelling catheters are more often than not the origin, which makes these infections potentially preventable with vigilance. In 2011, Clinical Infectious Diseases published guidelines on the prevention of catheter associated infections. The article contends that the etiology of infection commonly comes from migration of skin flora or direct contamination, which can be avoided with sterile techniques. Recommendations highlight the importance of faculty and staff education on hand hygiene, proper aseptic techniques, and surveillance [18]. Line placement is also important in infection prevention. According to the CDC, placement should be guided by ultrasound when faculty are trained appropriately, and femoral lines should be avoided whenever possible given the high rate of infection at that site. Subclavian is always preferable for central venous lines, and the least number of ports needed should be used. Central lines need not be changed unless there is high suspicion for infection as frequent manipulation is associated with higher rates of infection. Peripheral lines, however, should be exchanged every 72–96 h [19]. It is estimated that 84% of primary BSI is associated with central venous lines [7].

Of course, there are times when preventive measures fail, and infection occurs. It is essential, then, that infection is rapidly identified and treated appropriately without delay. Though BSI’s can be caused by Gram-positive bacteria, Gram-negative infections are on the rise. Common pathogens include *E. coli*, *K. pneumoniae*, *A. baumannii*, and *P. aeruginosa*. A one- year prospective study at a tertiary referral center in Brazil sought to determine independent risk factors for the development of resistance among these Gram-negative infections with the goal of providing effective therapy without delay. Investigators found that 65% of BSI were hospital acquired and 28.7% were multi-drug resistant (MDR). Though there was some variety among pathogens, 80% were *Enterobacteriaceae* spp. with *K. pneumoniae* positively associated with MDR (*p* < 0.05). Greater than 50% of the *K. pneumoniae* isolates were resistant to ampicillin-sulbactam, piperacillin-tazobactam, ceftriaxone, ceftazidime, cefepime, and ciprofloxacin. *A. baumannii* was also associated with high prevalence of resistance, but interestingly, all isolates were susceptible to tigecycline. Risk factors for developing resistance in this study were as follows: male sex, age >60, previous use of antibiotics (particularly fluoroquinolones), and the presence of liver disease. Patients with liver disease were 4.9 times more likely to acquire or develop MDR [13]. While this study was limited geographically to Brazil, these risk factors are consistent with other literature on the subject of multi-drug resistance. Another study evaluating the risk factors associated with hospital acquired *A. baumannii* BSI found that antibiotic use prior to diagnosis, SAPS II scores, and age were also independent risk factors for development of resistance. This was especially true for previous use of carbapenems (OR 11.96) and fluoroquinolones (6.71) [20]. Awareness of the characteristics predisposed to resistance allows practitioners to tailor empiric therapy according to local trends in pathology and resistance. Final therapy, as always, should be guided by culture and sensitivity data. 

When risk factors for resistance are not clearly present, there has been debate on whether to start monotherapy or combination therapy empirically. A three-year study in South Carolina examined the 28 day all-cause mortality rate in patients with monomicrobial Gram-negative BSI who received a β lactam compared to those who received a β-lactam in combination with either fluoroquinolone or aminoglycoside antibiotics. The study ultimately revealed no difference in 28-day mortality. Empiric treatment with β-lactam monotherapy was found to be appropriate for >90% of isolates, with the most common pathogen identified as *Escherichia coli*. Therefore, in patients without identifiable risk factors for resistance, there is no need for combination therapy, and there should be more judicious use of antimicrobials outside of βlactams [21].

### 3.3. Intra-Abdominal Infections

Intra-abdominal infections (IAI) are infections of the peritoneal space, which can be defined by abscess formation or diffuse peritonitis. They can vary greatly in severity, but are universally associated with high rates of morbidity and mortality given that the most vulnerable population are critically ill patients [22]. There are three categories of IAI: primary, secondary, and tertiary. Primary infections are the least concerning and are typically localized. Outpatient treatment with antibiotics is typically sufficient. Secondary infections are most common and occur as a result of the breakdown of surgical anastomoses, perforation, traumatic injury, or ischemic necrosis. Tertiary infections are hospital acquired and generally occur post-operatively [23]. The two main tenets of IAI treatment are source control, via surgical or percutaneous drainage, and appropriate antimicrobial coverage. Severe infections can present with signs and symptoms of diffuse peritonitis or even shock, in which case, treatment should be initiated immediately with broad spectrum empiric antibiotics and surgical intervention for adequate source control. While mild, community acquired, IAI with no signs of hemodynamic instability can be treated empirically, healthcare-associated infections or infections in patients with increased risk of treatment failure should always be guided by culture and sensitivity data [24]. The need for drainage of infected fluid gives practitioners the unique opportunity to obtain cultures for the majority of these infections. This microbiologic data becomes invaluable in studying MDR pathogens and the development of new antimicrobial therapies. 

Deciding on empiric treatment for these infections can be challenging given the wide variety of possible pathogens. It is imperative to determine the source of infection from the history, physical, and imaging modalities and guide therapy accordingly. The pathogen responsible for an infection of biliary tree, for example, can be very different from that of appendix. Table 1 and Table 2 below depict the common pathogens per anatomic region. Initial treatment should follow the Tarragona strategy. Begin with high dose, broad spectrum antibiotic tailored to whether the patient and the hospital are at a high risk for resistance. Empiric treatment should cover enteric Gram-negative bacteria. Obligate anaerobe coverage should be included if the source is small bowel, appendix, or colon [24]. When culture and sensitivity data are available, de-escalate, and narrow coverage [23].

### 3.4. Urinary Tract Infections

Urinary tract infections (UTI) are extremely prevalent both in the community and healthcare settings. Infections within the urinary tract can be divided into lower, cystitis, and upper, pyelonephritis, infections. They can be further categorized as complicated or uncomplicated. Uncomplicated infections are defined as cystitis in non-pregnant, premenopausal females with no health conditions or anatomic variations within the urinary tract. These infections are generally clinically diagnosed, and treatment is usually empiric [25]. Symptoms associated with UTI include dysuria, increased frequency and urgency of urination, flank pain, suprapubic tenderness, hematuria, CVA tenderness, and fever. The co-existence of dysuria and increased urinary frequency in females in the absence of vaginal symptoms increases the likelihood of correct, uncomplicated acute cystitis diagnosis by 90%, thereby practically negating the need for laboratory testing. When additional testing is warranted, a urine dipstick suffices in the case of uncomplicated UTI as the absence of leukocyte esterase has an NPV greater than 90% [25,26]. Pathogens most commonly isolated from uncomplicated cystitis are *Escherichia coli*, *Klebsiella pneumoniae*, *Proteus mirabilis*, and *Staphylococcus saprophyticus.* Empiric treatment should provide coverage as such. First-line antibiotics should be nitrofurantoin 100 mg twice daily for a duration of five days. Trimethoprim sulfamethoxazole (TMP-SMX) 160 mg/800 mg twice daily for three days is an acceptable alternative, but only when local resistance is less than 20%. Unfortunately, with the over-prescription of TMP-SMX, we are starting to see widespread resistance, and the use of nitrofurantoin is preferred. While fluoroquinolones are also an effective alternative, given their propensity for toxicity and subsequent resistance, the IDSA recommends reserving use for upper urinary tract infections. Pyelonephritis can be empirically treated with 500 mg of ciprofloxacin twice daily for seven days, though cultures and sensitivities should always be performed for such infections [27]. Additionally, microbiologic data should be collected for any serious infection or when there are known risk factors for antimicrobial resistance. Serious infections include, but are not limited to, those with signs of hydronephrosis or associated with trauma, use of urinary catheters, drug resistant pathogens, or cirrhotic liver. Risk factors for resistance include age over 60, use of an indwelling catheter, travel outside of the country within the last 3–6 months, chronic medical conditions, and history of UTI or antibiotic resistance [28].

Catheter associated urinary tract infection (CAUTI) is the most common hospital acquired infection in the U.S. and, as such, is more commonly associated with antimicrobial resistant pathogens. CAUTI is defined as urinary tract infection symptoms with no alternative likely cause and greater than 10^3^ cfu/mL of urine in a patient whose catheter had been removed within the previous 48 h. Unfortunately, much of catheter use is inappropriate, and our greatest defense against these infections is to limit use. Treatment should always be tailored to urine culture and sensitivity data. Duration is typically seven days, though extension up to 14 days may be necessary if symptoms persist [28]. 

### 3.5. Therapeutic Approaches to Gram-Negative Infections 

Over the last decade, there has been an initiative for the research and development of new antimicrobial options for the growing problem of multi-drug resistant organisms. In 2015, The National Action Plan was published, which described the U.S. National Strategy for Combating Antibiotic-Resistant Bacteria. A five-year plan was created to increase research and innovation in the development of new antimicrobials and surveillance techniques to better inform the medical and veterinary communities about trends in resistance [29]. While many strides have been made in surveillance strategies, antimicrobial stewardship, and prevention, few developments have emerged in terms of antimicrobial agents. As such, there has been a resurgence of and increased reliance on older antibiotics, which had until late fallen out of fashion. Here, the character and use of both old and new antibiotics will be detailed. 

### 3.6. The Resurgence of Old Antibiotics

#### 3.6.1. Colistin 

Colistin is a powerful polymyxin developed in the 1950s with potent activity against *Escherichia coli*, *Klebsiella pneumoniae*, *Pseudomonas aeruginosa*, and *Acinetobacter baumannii.* Its use has fallen out of favor over the last decade because of its systemic toxic effects such as nephrotoxicity and neurotoxicity [30]. With resistance rising though, physicians are having to revisit this powerful drug to treat complicated urinary tract infection (cUTI), pyelonephritis, BSI, IAI, and VAP. The mechanism through which it works is electrostatic destabilization of the outer membrane of Gram-negative bacteria. Colistin disrupts the divalent bonds within LPS structures, leading to the leaked contents of the bacterial cell and subsequent cell death [31]. It is generally used in combination with carbapenems such as meropenem to combat resistance, though monotherapy for patients with low risk BSI is acceptable [21]. 

Unfortunately, with increased frequency of use in medicine and continued use in veterinary medicine as a growth factor, resistance is starting to emerge among *Enterobacteriaceae* spp., *Pseudomonas aeruginosa*, and *Acinetobacter baumannii.* Chromosomal mutations in plasmid gene mcr-1 are largely to blame for resistance, and evidence for this gene has been observed in water systems, manure, and city drainage. Additionally, modification of the Lipid A structure through the gene bfmr has been implicated in colistin resistance [8]. A fecal study of livestock animals in Poland, Taiwan, China, and Switzerland found that colistin resistance is highest among animals. Resistance can easily spread to humans through ingestion of fecal contamination and horizontal gene transfer among host bacteria [32].

It appears there may also be a relationship between carbapenem resistance and the development of colistin resistance. The European Antimicrobial Resistance Surveillance Network (EARS-net) reported that 29% of carbapenem resistant *K. pneumoniae* were also resistant to colistin, while only 3% of carbapenem susceptible *K. pneumoniae* were colistin resistant (CoR). Similar results were found in a retrospective study in Dubai in which 27% of carbapenem resistant *K. pneumoniae* were also resistant to colistin in five of their major hospitals [32]. 

Although this rise in resistance is of grave concern, colistin is still one of our most potent defenses against MDR Gram-negative bacteria. A retrospective case-control study of ESBL Gram-negative infections in a tertiary care center in Australia found that all New Delhi metallo-β-lactamase 1 (NDM-1) producing pathogens were resistant to all other antibiotics except colistin [11]. Colistin remains strongly active against MDR *P. aeruginosa* and *A. baumannii. A. baumannii* is an infection of particular concern given its propensity to develop in critically ill patients in the hospitalized setting. It is associated with high morbidity, mortality, and increased healthcare costs. A Brazilian study examining the synergist effects of sulbactam with various antimicrobials against A. baumannii found that all 30 isolates were susceptible to colistin alone, and 43% of isolates saw synergy between sulbactam and colistin followed only by 27% of isolates that were susceptible to tigecycline-sulbactam [33]. Although colistin remains a relatively reliable weapon against MDR Gram-negative bacteria, with the rise in resistance and its toxic systemic effects, it is imperative that this drug is reserved as a last defense. 

#### 3.6.2. Fosfomycin

This antibiotic was developed in 1969 and works by inhibiting the initiation of peptidoglycan synthesis in Gram-positive and Gram-negative organisms. Although its mechanism of action is similar to that of penicillin by preventing cell wall synthesis, its method in doing so is unique, which means relatively low chances of resistance development. It works well against MDR *Pseudomonas aeruginosa* and ESBL *Klebsiella pneumoniae* and has excellent lung and urinary penetration. A systematic review found that 90% of MDR strains of *P. aeruginosa* were susceptible, and a study of 5057 ESBL producing isolates of E. coli and K. pneumoniae found that 91.3% were susceptible [30,34,35]. With high levels of clinical efficacy and low levels of resistance, it is seemingly ideal for GNB infections. In fact, IDSA recommends use for simple cystitis given that *E. coli* resistance is rising due to overuse of antibiotics such as ciprofloxacin and TMP-SMX in outpatient offices. The mechanism of resistance is most often AmpC βlactamase, which is carried within plasmids and transferred horizontally. Fosfomycin and nitrofurantoin are recommended treatments for this type of resistance as they have been found to be active against 90% of ESBL organisms in lower urinary tract infections [5,25]. Furthermore, recent studies have shown fosfomycin to be noninferior to piperacillin-tazobactam in cUTI and pyelonephritis with 80% of ESBL *Escherichia coli* and *Klebsiella pneumoniae* isolates susceptible [34]. Unfortunately, though it is widely used in other countries, its availability in the United States is sparse. 

#### 3.6.3. New Antibiotics

A variety of new drugs and new drug combinations have been developed in recent years to fight MDR gram- negative infections. This section will focus on each new antimicrobial as it pertains to mechanism of action, target bacteria and current FDA approvals. A summary of these therapies can be found below in Table 2.

#### 3.6.4. Plazomicin 

Plazomicin is a synthetic aminoglycoside that works by inhibiting protein synthesis as is the case with all aminoglycosides. It differs, however, by maintaining stability even against aminoglycoside modifying enzymes. It has activity against MDR *Enterobacteriaceae.* Resistance occurs with rRNA methyltransferases in *Enterobacteriaceae*, *Pseudomonas aeruginosa*, and *Acinetobacter baumannii*. In vitro study of 300 carbapenem resistant *Enterobacteriaceae and K. pneumoniae* isolates showed 87% susceptibility. Additionally, 79% of *A. baumannii* were susceptible [36].

The EPIC trial (Evaluating Plazomicin in cUTI) was an international phase 3 randomized controlled trial examining the efficacy of plazomicin once daily for the treatment of cUTI versus standard therapy of meropenem. It was determined to be non-inferior within a 10% margin of clinical cure rate. At the test-of-cure visit, the plazomicin group’s clinical cure rate, in fact, remained higher than patients treated with meropenem. It was subsequently approved for cUTI and pyelonephritis with *E. coli*, *K. pneumoniae*, and *P. mirabilis* [36]. Pharmacokinetic studies have also shown excellent lung penetration, so there is potential for VAP treatment in combination with β-lactamase inhibitor combinations [5,36].

*Acinetobacter baumannii* is one of the most dangerous threats in the world of antibiotic resistance as it has become increasingly resistant to antibiotics such as aminoglycosides and carbapenems, which were previously mainstays of treatment. An in vitro study of MDR *A. baumannii* found that plazomicin was more potent against carbapenem resistant *A. baumannii* than amikacin or gentamicin even with all isolates carrying aminoglycoside modifying enzymes. Combination therapy with plazomicin and imipenem or meropenem proved synergistic in all isolates but one [37]. These are promising results when considering how deadly MDR *A. Baumannii* can be with dwindling therapeutic options available. Adverse events are important to consider, though, with toxicities such as nephrotoxicity and ototoxicity.

#### 3.6.5. Tigecycline 

Tigecycline is a glycylcycline with broad spectrum Gram-negative activity used as a salvage therapy for ESBL and carbapenem resistant *Enterobacteriaceae* and *Acinetobacter baumannii* infections. It does not have anti-pseudomonal properties and has low urinary concentration [34]. It works through inhibition of protein synthesis by binding to the 30S ribosomal subunit. A study of serious infections including cIAI, pneumonia, SSI, and bacteremia due to MDR Gram-negative bacteria including *A. baumannii*, *K. pneumoniae*, and *E. coli*, found that IV tigecycline provided 72.2% clinical cure rates and complete bacterial eradication in 66.7% of patients [5].

It was approved for use in patients with cIAI and cSSI in 2006 and CAP in 2009. A 2011 surveillance study of 22,005 isolates of meropenem resistant *K. pneumoniae* and *A. baumannii* found that tigecycline susceptibility was near 100%, but resistance is on the rise, with estimates now nearly 50% of isolates non-susceptible. A large study on antimicrobial efficacy against MDR *A. baumannii* found that susceptibility to tigecycline alone was 47%. It was found to be synergistic with sulbactam with 27% of isolates responding better to the combination [30]. Efflux pumps are responsible for a large portion of this resistance. AdeABC is the superfamily of efflux pumps responsible for aminoglycoside resistance. Gene targeted therapy for the gene Ade-R, which regulates AdeABC, was examined as a possible solution to tigecycline resistant *A. baumannii*; however, the results were not promising, indicating that Gram-negative bacteria like *A. baumannii* likely have multiple mechanisms of resistance to tigecycline [38]. Use is now falling out of favor while cure rates are dropping in comparison with carbapenems such as imipenem, as well as a higher mortality rate when compared to colistin [17]. Tigecycline is still recommended, however, in treatment of high-risk patients with secondary peritonitis and a high risk of resistance [23].

#### 3.6.6. Ceftolozane-Tazobactam 

This new cephalosporin and β-lactamase inhibitor combination was approved for the treatment of cIAI, cUTI, HAP, and VAP as of June 2019. It is a powerful anti-pseudomonal cephalosporin paired with a well-established β-lactamase inhibitor, tazobactam. Ceftolozane is part of the oxyimino subset of cephalosporins. It is difficult for resistance to build as this combination is unaffected by AmpC overproduction, efflux pumps, or porin alteration, which are major mechanisms of resistance for ESBL producing organisms. Other coverage includes MDR *Enterobacteriaceae* with wide coverage across the Ambler Classes of ESBL producing pathogens [7]. A recent study of 53 patients in the ICU with VAP secondary to *P. aeruginosa* infection showed that the ceftolozane-tazobactam (C/T) combination was more potent than imipenem, ciprofloxacin, or ceftazidime-avibactam [34]. In a European population-based surveillance study, C/T was found to be the most effective β-lactam against *P. aeruginosa* even to those non-susceptible to cefepime, meropenem, or piperacillin-tazobactam. Against *Enterobacteriaceae* spp., it was second only to meropenem in efficacy [39]. As this is the most potent drug in the arsenal of MDR *P. aeruginosa*, it should become the standard empiric therapy for those patients at high risk for MDR VAP except when it is known that the strain of *P. aeruginosa* produces carbapenemases [34]. 

Other studies have evaluated the use of C/T in cUTI and cIAI. The multinational, double-blinded phase 3, non-inferiority trial ASPECT found C/T to be non-inferior in clinical cure rate and mortality for cUTI or pyelonephritis to high-dose levofloxacin when both were given for seven days [5]. Another phase 2, double-blinded, randomized control trial examined the efficacy of C/T in patients with cIAI in comparison with meropenem. Patients with cIAI secondary to *E. coli* treated with C/T plus metronidazole achieved a clinical cure rate of 89.5%, while patients with *K. pneumoniae* had a cure rate of 100% [5,7]. 

#### 3.6.7. Aztreonam

Aztreonam is a monobactam with activity against metallo-β-lactamase, which has recently been paired with a novel diazabicyclooctane (DBO) β-lactamase inhibitor, avibactam. The combination provides adequate coverage of Ambler Classes A, B, C, and some D with limited coverage of *A. baumannii* and *P. aeruginosa* [22].

In 2017, Pfizer completed an international, open-label phase 2 prospective study titled REJUVENATE on the pharmacokinetics and safety of the use of aztreonam paired with novel β-lactamase inhibitor avibactam versus aztreonam alone in patients hospitalized with cIAI. Adverse events were similar between the two groups, demonstrating the safety of pairing the two [40].

Pfizer began another study examining aztreonam-avibactam with metronidazole versus meropenem-colistin for the treatment of severe MDR Gram-negative infections, but enrollment has been temporarily suspended due to the lack of investigational product availability [40,41,42,43]. The FDA recently granted Fast Track designation to the drug combination given its potential to treat deadly, resistant cIAI and cUTI infections [44].

#### 3.6.8. Ceftazidime-Avibactam 

Another novel cephalosporin-β-lactamase inhibitor combination to hit the market recently is ceftazidime-avibactam. Avibactam is part of a new class of β-lactamase inhibitors called diazabicyclooctanes (DBO), which work by reversibly binding to β-lactamase enzymes, allowing for the recycling and rebinding of extended-spectrum β-lactamases, which restores the activity of ceftazidime potentially over 1000-fold. The spectrum of its coverage ranges through Ambler Classes A (ESBL, KPC), C (AmpC), and D (OXA-48) [7,45]. It has been approved for treatment of cIAI (when paired with metronidazole), cUTI, and pyelonephritis. A phase 2 double-blinded, randomized controlled trial found that ceftazidime-avibactam plus metronidazole was non-inferior to meropenem in patients with cIAI due to *E. coli* with clinical cure rates of 91.2% in the C/A plus metronidazole group and 93.4% in the meropenem group. Even in groups with pathogens non-susceptible to ceftazidime, clinical cure rates exceeded 94% [5]. Another study examined the in vitro efficacy of C/A against isolates *of P. aeruginosa* and *Enterobacteriaceae* spp. Results showed inhibition of 99% of 36,380 isolates of *Enterobacteriaceae* spp. This included 99.2% of MDR isolates and 97.8% of XDR isolates. Additionally, 97.1% of *P. aeruginosa* isolates were inhibited, which, too, included MDR and XDR strains [45]. An international 2013 open-label phase 3 clinical trial, REPRISE, evaluated the use of C/A in the treatment of cUTI and cIAI with ceftazidime resistant *Enterobacteriaceae* and *P. aeruginosa*, revealing that the combination yielded similar clinical cure rates to those of the best available, carbapenem [25].

Ceftazidime-avibactam has been approved for HAP and VAP in Europe for several years, but only in 2018 received approval for the same in the United States. This was the first new antibiotic approved for the treatment of Gram-negative pneumonia infections in over a decade. REPROVE was a phase 3 trial evaluating the efficacy of C/A as compared to meropenem in the treatment of HAP and VAP in the U.S. The most common pathogens were *Enterobacteriaceae* spp. and *P. aeruginosa.* C/A was found to be non-inferior to meropenem in both clinical cure rate and 28-day mortality [46].

#### 3.6.9. Imipenem-Colistatin Plus Relebactam 

Relebactam, similar to vaborbactam, is a new generation of β-lactamase inhibitor designed to treat carbaepenemase producing Gram-negative bacterial infections [47]. Investigations have paired it with the powerful carbapenem and imipenem given their compatible pharmacokinetic properties. Relebactam has been shown to restore the activity of imipenem in treating *K. pneumoniae* and *P. aeruginosa* by reducing the MIC by 32-fold. Phase 2 non-inferiority trials have revealed imipenem-relebactam to be similar in efficacy to imipenem-cilastatin alone in treating cIAI and cUTIs. In fact, clinical cures were 95% and 97%, respectively, for 351 patients with cIAI treated with imipenem-relebactam or imipenem-cilastatin [47].

RESTORE-IM1 was a phase 3, RCT DB evaluating the efficacy of imipenem-relebactam when compared to colistin-imipenem for imipenem non-susceptible infections in patients with HAP/VAP, cUTI, or cIAI. The most common pathogens identified were (77%) *P. aeruginosa* and (16%) *K. pneumoniae* with AmpC, ESBL, KPC, and OXA β-lactamases. Clinical response rates were nearly identical in both arms with 71.4% in the imipenem-relebactam arm and 70.0% in the imipenem-colistatin arm. The best clinical response was observed in patients receiving imipenem-relebactam with HAP or VAP. A stark difference was seen in treating *P. aeruginosa* with 81% of imipenem-relebactam patients having a favorable response versus only 63% of imipenem-colistatin patients. Additionally, all-cause mortality was 20% lower in patients receiving imipenem-relebactam, and there were significantly less nephrotoxic events compared to imipenem-colistatin [22,48].

#### 3.6.10. Cefiderocol

Cefiderocol is a unique, novel antibiotic with broad spectrum Gram-negative bacterial coverage. It is the first antibiotic in its class of siderophore cephalosporins. It functions as a trojan horse, gaining entrance to bacterial periplasm bound to iron needed for cell survival. The development was inspired by the siderophore machinery of the bacteria itself, which extends into the host environment to obtain iron and transports it back inside the cell beyond the protective outer membrane [22,40,49]. This is an exciting development as there is potential to use antibiotics that previously could not penetrate the outer membrane of Gram-negative bacteria. One study was able to use daptomycin as a potent inhibitor of in vitro and in vivo *A. baumannii* by mixing it with a synthetic analog of A. *baumannii*’s own siderophore [49]. In November of 2019, cefiderocol gained its first approval for the treatment of cUTI or pyelonephritis in adults with little to no other treatment options. The future for this type of antibiotic is promising and may hold a key in the fight against resistance.

#### 3.6.11. Eravacycline

This is a fluorocycline antibiotic that works by inhibiting the 30S ribosomal subunit with broad spectrum coverage of extended-spectrum β-lactamases, carbapenem resistant *Enterobacteriaceae*, MRSA, *A. baumannii*, and vancomycin resistant *Enterobacteriaceae*. In vitro studies have shown eravacycline to be a potent inhibitor of MDR *E. coli* with MIC <0.5 mg/L, and it has since been approved for the treatment of cUTI. Generally, it is well tolerated with the main side effect being nausea [7].

The IGNITE-1 trial was a phase 3, double-blinded, randomized controlled trial evaluating 1 mg/kg of eravacycline intravenously every 12 h versus 1g of ertapenem every 24 h for four days that found that eravacycline was non-inferior to ertapenem in treating patients hospitalized with cIAI requiring percutaneous drainage or operative intervention with clinical cure rates of 87% and 89% [34,40]. This study led to the approval of eravacycline for cIAI in 2018. 

In vitro eravacycline studies have also shown promising efficacy against carbapenem resistant *A. baumannii* (CRAB) infections. In a study of CRAB infections with known Ambler Class B and/or Class D enzymes, eravacycline was found to be the most potent when compared to aminoglycosides, βlactams, colistin, tetracyclines, and fluoroquinolones. Unfortunately, the same success was not observed in in vivo studies, so optimism should be cautious for the future of treating *A. baumannii* infections [34].

In healthy adults, a pharmacokinetic study of IV eravacycline also revealed penetration of the pleural lining 50-fold higher than that of plasma, indicating the potential for use in HAP and VAP [5].

#### 3.6.12. Meropenem-Vaborbactam

Carbapenem resistance is growing world-wide, which is of grave concern as carbapenems were previously our strongest and broadest antibiotics for otherwise resistant *Enterobacteriaceae* spp., *P. aeruginosa*, and *A. baumannii.* The World Health Organization (WHO) ranks these carbapenem resistant bacteria among the highest threats to humanity and a top priority in terms of research and development of new antibiotics. Vaborbactam was developed specifically to fight carbapenemase producing pathogens. It belongs to a new class of βlactamase inhibitors and restores the activity of meropenem against carbapenem resistant *Enterobacteriaceae* with KPC production. It was approved in 2017 by the FDA. It is effective against Ambler Classes A and C; however, it does not work against Ambler Class B *P. aeruginosa*, *A. baumannii*, or *S. maltophilia* [24]. The trial TANGO I showed the combination to be non-inferior to piperacillin-tazobactam in treating cUTI. It was approved for cUTI, pyelonephritis, cIAI, HAP, VAP, and BSI [34]. TANGO II was a randomized controlled trial testing the efficacy of M/V against best available treatment carbapenem resistant *Enterobacteriaceae* HAP/VAP. It showed higher clinical cure rate in carbapenem resistant *Enterobacteriaceae* HAP/VAP when compared to the best available treatment [47]. 

### 3.7. Other Therapeutic Approaches 

With so few new antimicrobials on the horizon, scientists have started to evaluate other ways in which to combat antibiotic resistance. One area of successful research is in antimicrobial peptides (AMPs). These peptides exist as immune molecules in nature and have powerful antimicrobial activity against Gram-positive and Gram-negative bacteria, fungi, and parasites. Unfortunately, with thousands of these molecules known to exist, research is time-consuming and expensive, which has hindered progress. A study evaluating the in vitro activity of 14 animal derived AMPs against MDR and XDR *A. baumannii*, MRSA, ESBL *P. aeruginosa*, and ESBL *E. coli* had encouraging results. Two AMPs, cathelicidin-BF and tachyplesin III, were found to be effective at killing all strains of MDR *A. baumannii.* There is speculation that the secondary and tertiary structures of these molecules determine their lethality, but interestingly, the two AMPs noted differ in both secondary structure and hydrophilic/hydrophobic arrangements [50]. An in vivo study examining tachyplesin III’s role in combating concomitant *P. aeruginosa* and *A. baumannii* lung infections found that pre-treatment with the peptide reduced inflammation, bacterial burden, and lung damage by increasing phagocytosis and reducing cytokine release in mice [51]. If funding can be secured, AMPs may be the future of antimicrobial treatment strategies. 

Another area of promising research is actually a resurgence of an older antimicrobial treatment strategy: bacteriophage therapy. Prior to the advent of penicillin in the 1940s, phage therapy was used to treat bacterial infections. Bacteriophages are found in virtually all environments. They are extremely diverse and highly specific to their hosts, and lytic phages are very effective at killing a wide variety of bacteria. There are a number of advantages to using phage therapies when compared to antibiotics. Their high specificity means that bacteria can be precisely targeted with little to no damage to surrounding tissues, and systemic side effects are uncommon. As the phages are viruses, they are capable of evolving with their hosts, evading resistance mechanisms they develop and preventing future resistance mechanisms by reducing selective pressure. Phages can be used in a number of applications, most notably in treating bacteria that form biofilms. There are a variety of methods using phages that have been shown to be effective in in vitro and some in vivo studies. The more successful applications have been in antibiotic-phage combinations and phage cocktails. Combining phages with antibiotics reduces time-to-cure and the dose required to treat, thereby reducing the harm of the antimicrobial and bacterial components. Phage cocktails further reduce the threat of resistance. The PEV20 phage has been shown to be in synergy in treating *Pseudomonas aeruginosa* when combined with ciprofloxacin, and phage T4 when combined with cefotaxime has been effective against biofilm formation and *Escherichia coli*. Unfortunately, there is limited data on in vivo studies currently, which has stalled progress in phage therapy, and as with all treatment, there are some disadvantages to using phages. For example, the pharmacokinetics and pharmacodynamics of developing phage therapies are complicated. Determining proper dosing and therapy duration will require much more dedicated research, and guidelines for safety need to be developed [52,53].

Gene editing has potential for growth as well. As we continue to identify specific mechanisms of Gram-negative bacterial resistance and the genes responsible for those mechanisms, there is potential to edit and knock-out those genes. An experiment targeting Ade-R, the regulator of AdeABC-mediated efflux transcription in MDR *A. baumannii*, tried to do just that. Although the knock-out of Ade-R was successful, the isolates still remained largely multi-drug resistant, indicating other mechanisms of resistance likely exist. The research still showed that this kind of genome editing can be used to study and manipulate MDR and XDR bacteria [38]. 

Targeted drug delivery, which is more often researched in the development of chemotherapeutic cancer agents, should also be considered for antimicrobial treatment. Targeted delivery reduces side effects and dosing requirements, making it an ideal system. The complexity of the chemistry and human physiology has limited our options, but nanosponges may be the answer. These structures are roughly the size of a virus and composed of a stable, hollow, spherical structure, which can be loaded with a variety of drug types. Their size and stability in lipophilic and hydrophilic environments open the possibility of creating oral, parental, and topical preparations [54].

Other areas of interest include sterilization techniques such as nanosecond electrical pulses (EPs), visible light therapy (VLT), and photothermal therapy (PTT). These strategies are particularly helpful in the treatment of topical infections and sterilization of indwelling lines and catheters. EPs work in conjunction with antimicrobials by disrupting the bacterial membrane, allowing for more concentrated doses to enter the cell. This reduces treatment time and dose. In a study assessing the synergy between antimicrobials, tobramycin, and rifampicin and variable EPs in treating *Staphylococcus aureus* and *Escherichia coli*, 300 ns of EPs enhanced the activity of rifampicin. Complete sterilization of *Escherichia coli*, as defined by a nine-log colony reduction, was achieved with 20 μg/mL of rifampicin and 445 ns of a 30 kV/cm field, which represented a four-log reduction improvement upon 20 μg/mL of rifampicin alone [55]. Visible light therapy is currently FDA approved for use for acne vulgaris, and research is being done to determine its role in treating SSI and device associated Gram-negative infections. An in vitro study evaluating the efficacy of violet 405nm light in treating ampicillin resistant *Escherichia coli* found an 81.7% reduction of bacterial growth with irradiance of 2.89 mW/cm^2^ over 120 min (*p* < 0.001) [56]. Similarly, a review of both Gram-positive and Gram-negative bacteria susceptibility to violet light therapy between 380 and 480 nm found a 91% inactivation of *Pseudomonas aeruginosa* with 405 nm of VLT [57]. Finally, heat has been shown to be effective in disrupting the microbial membrane and therefore killing bacteria. A variety of inorganic materials such as ultrasound and microwaves can be used to concentrate a heat source onto a probe aimed at a treatment area. An in vivo study showed efficacy in treating rats experimentally infected with resistant strains of *Escherichia coli*, *Acinetobacter baumannii*, and *Enterococcus* spp. A thermosensitive polymer, n-vinyl polycaprolactam (PVCL), was used as a gel medium to disperse gold nanorods excited by a laser. *Acinetobacter baumannii* and *Escherichia coli* saw a 94% and 96% viability reduction with 40 min of laser exposure (*p* < 0.001). Furthermore, there was no difference in the surrounding tissues in rats with punch biopsy only versus treated rats, indicating the safety of this method [58].

### 3.8. Antibiotic Stewardship 

As we look forward in the fight against antimicrobial resistance, we must consider proper antibiotic stewardship in line with the development of new drugs in our defense. The concept of “antibiotic stewardship” was first used in the late 1980s when the Infectious Diseases Society of America (IDSA) created guidelines for the more judicious use of antibiotics to reduce the ever-growing concern of resistance. Despite the guideline though, for many years, stewardship practices varied widely from institution to institution. It was not until 2007, when the CDC acknowledged a critical point had been reached for antibiotic resistance worldwide. The IDSA then developed a guideline on the creation and implementation of antibiotic stewardship programs (ASP) for inpatient, acute care hospitals. In 2014, the CDC recommended that all U.S. hospitals have a program in place [59,60,61].

The CDC outlines “core elements” of an effective antibiotic stewardship program as defined by IDSA. A proper ASP includes a leadership commitment by the institution to provide financial, technological, and human support, a program director, ideally a physician trained in infectious disease, a leading pharmacist, implementation of actions, a way of tracking improvement, and finally, a reporting system for all relevant staff. As of 2016, only 64.2% of hospitals across the U.S. have met all criteria. The goal is to get to 100% by 2020 [31,60]. In 2016, the Healthcare Infection Control Practices Advisory Committee (HICPAC) released recommendations to further guide the implementation of ASPs. They included principles of testing, which encourage expeditious diagnosis, culture, and susceptibility testing, as well as principles of treatment, which highlight the importance of appropriate empirical treatment, adequate source control, proper dosing, duration of therapy, and de-escalation of therapy whenever possible [15,61].

There has been a multitude of studies showcasing the efficacy and success of ASPs around the world. A thirteen-year observation study performed at a large tertiary care teaching hospital in North Carolina sought to determine the long-term effects of implementing an ASP. The results revealed sustained reductions in antimicrobial use and resistance. Over the study period, total antibiotic use was reduced by 62.8% (*p* < 0.0001). Additionally, there was a 56.8% reduction in MRSA infection [62].

A multicenter retrospective study assessing rapid diagnostic testing with matrix assisted laser desorption/ionization time-of-flight mass spectrometry (MALDI-TOF) in conjunction with an established ASP found that they were able to reduce mean identification time after culture from 32 h (+/− 16 h) to 6.5 h (+/− 5.4 h) (*p* < 0.001), which in turn led to reduced time to therapy adjustment (48 +/− 22 h to 23 +/− 14 h, *p* < 0.001) and a reduction of $3411.00 less per patient in the intervention group [63].

A similar, single-center retrospective study assessing the use of MALDI-TOF in conjunction with an ASP for the diagnosis and treatment of pneumonia or bacteremia with *Acinetobacter baumannii* also found a significant reduction in time to appropriate therapy and length of stay, as well as a 19% increase in clinical cure rate at seven days (*p* = 0.016) [64].

Outside the controlled arena of healthcare systems, there are multiple other factors contributing to antimicrobial resistance. In both industrialized and developing countries, the use of antibiotics in agriculture is often excessive and inappropriate. Globally, antimicrobials dispensed to animals far exceed those to humans. In the United States alone, 80% of antibiotics sold are distributed for use in animals with 70% of those being clinically relevant in human medicine [65,66]. Although policy exists in most countries to limit unnecessary use in agriculture, these laws are rarely strictly enforced. This is particularly true of developing countries under pressure to produce and export increasing amounts of animal products, which encourages the use of antibiotics as growth promoters. As these drugs are administered at subinhibitory levels, bacteria have the advantage of evolving resistance. Additionally, antibiotics are often used prophylactically to prevent infection in herds, which is ultimately detrimental to animals and human consumers [67].

While the exact mechanisms of animal to human transmission have yet to be elucidated, it is well understood that antimicrobial resistance in animals and agriculture leads to increased human exposure to antibiotic resistant bacteria through both direct interaction with animals and indirectly through the consumption of contaminated animal products, vegetation, and water supply. Farmers and primary food processing workers are exposed to animal feces and blood, which naturally increases the risk of antimicrobial resistant gene transmission. An estimated 58% of veterinary antibiotics also end up in our ground water and ruffage through manure contamination of soil [63]. A microbiologic survey of 17 U.S. grazing cattle farms revealed some alarming trends in the spread of cefotaxime resistance. Fecal, soil, and plant samples were tested for cefotaxime resistant bacteria (CRB). Prevalence among cattle from all farms was 47.4%. Even more concerning, 95.7% of forage, 98.7% of soil, and 88.6% of water samples in the study had detectable levels of CRB. The farmers surveyed from these farms indicated that antibiotics were only used therapeutically under the supervision of a veterinarian [68].

Another major contributor to antibiotic resistance globally is the ease of access to and poor regulation of antibiotics in developing countries. In many parts of the world, antibiotics are readily available at local pharmacies, kiosks, and bodegas without prescription or counseling. As the understanding of antibiotics is poor among the general populations in these areas, people are often purchasing them for self-limited, viral illnesses, at inappropriate doses and durations. 

In 2010, the greatest users of antibiotics were India, followed by China and the United States. In fact, worldwide, there was a 76% increase in antibiotic use in the preceding decade. Twenty- three percent and 57% of that increase were from India and China, respectively [69]. A cross-sectional study of university students throughout six major regions of China found that only 38.7% of students understood that antibiotics did not work for viral illnesses, and 41.0% believed antibiotics sped influenza recovery. These students were also twice as likely as U.S. college students to request antibiotics for self-limited illnesses [70]. A similar study estimated that 75% of patients with suspected flu were treated with antibiotics yearly in China [71].

In countries such as Jordan, Nepal, and Indonesia, prescriptions are generally not required to obtain antibiotics. A four-month observational study in Jordan found that less than 70% of antibiotics dispensed were physician prescribed, and only 31.5% of those drugs were prescribed at the correct dosage and for the correct duration. Lower socioeconomic status and poor education were driving factors for self-medication, which was consistent with similar studies [72,73]. A cross-sectional, client simulation study in Surabaya, Indonesia, investigated common dispensation practices in the country. Over-the-counter (OTC) antibiotics were available without prescription at all pharmacies and 75% of roadside kiosks surveyed. Drugs were poorly labeled and packaged and often contained inadequate levels of active ingredients. Amoxicillin 500 mg, the most commonly dispensed drug, was found to contain only 45.8% active ingredients [74]. Clearly, there is a need for more education and policy change globally surrounding antibiotic dispensation practices. 

## 4. Conclusions

Resistance among deadly Gram-negative pathogens has risen to epidemic proportions, particularly within hospitals and acute care settings. Infections with bacteria such as MDR and XDR *Escherichia coli*, *Pseudomonas aeruginosa*, *Klebsiella pneumoniae*, and *Acinetobacter baumannii* contribute to alarmingly high rates of mortality in our most vulnerable populations and add billions of dollars in healthcare costs through lengthening hospital stays, utilization of resources, lost productivity, and high acuity care needs. 

Addressing this problem requires both infection prevention and appropriate treatment. Initiating evidence-based antibiotic stewardship programs will ensure better faculty and house staff education, appropriate diagnosis and treatment of disease, and ultimately, the reduction in the acquisition and spread of MDR bacteria. Knowledge of local patterns of resistance and individual risk factors for resistance will lead to better care of critically ill patients. 

Although there have been world-wide initiatives to develop new drugs for MDR Gram-negative pathogens, not much progress has been made over the last decade. We are forced to largely rely on new combinations of old drugs, and our most exciting advances have been with new β-lactamase inhibitors such as avibactam, vaborbactam, and relebactam paired with old cephalosporins and carbapenems. While these drugs effectively combat MDR pathogens, there is still a continued need for the development of new drugs and methods of combating resistance. For now, it seems our best offense is a good defense. 

## Figures and Tables

**Table 1 antibiotics-09-00196-t001:** Common bacterial pathogens stratified by anatomic region of source for empiric treatment reference [23].

Source of Infection	Common Pathogens
**Gastro-Duodenum**	*Streptococcus* spp., *E. coli*
**Small/Large Bowel**	*E. coli*, *K. pneumoniae*, *P. mirabilis*, *Bacteroides* spp., *Clostridium* spp., anaerobes
**Biliary Tree**	*Enterococcus* spp., *E. coli*, *K. pneumoniae*, *P. mirabilis*, *Bacteroides* spp., *Clostridium* spp.
**Appendix**	*E. coli*, *P. aeruginosa*, *Bacteroides* spp.
**Liver**	*Enterococcus* spp., *K. pneumoniae*, *E. coli*, *Bacteroides* spp.
**Spleen**	*Streptococcus* spp., *Staphylococci*
**Abscess**	*Enterococcus* spp., *E. coli*, *K. pneumoniae*, *Bacteroides* spp., *Clostridium* spp., anaerobes

**Table 2 antibiotics-09-00196-t002:** Emerging and recently established antimicrobials in the fight against Gram-negative antibiotic resistance. MDR, multi-drug resistant.

Antimicrobial Agent	Targets	Approvals
Plazomicin	MDR *E. coli*, *K. pneumoniae*, *P. mirabilis*, *A. baumannii*	cUTI and pyelonephritis
Tigecycline	ESBL, CR *Enterobacteriaceae*, *A. baumannii*	cIAI and cSSI
Ceftolozane-tazobactam	MDR *P. aeruginosa*, *Enterobacteriaceae* spp.	cIAI, cUTI, HAP, and VAP
Aztreonam-avibactam	ESBL *A baumannii*, *P. aeruginosa* (Ambler Class A-D)	cIAI
Ceftazidime-avibactam	MDR *Enterobacteriaceae* spp., *P. aeruginosa*	cIAI, pyelonephritis, cUTI, HAP, and VAP
Imipenem-colistatin-relebactam	MDR *K. pneumoniae*, *P. aeruginosa*	cUTI, cIAI
Cefiderocol	MDR, CR *P. aeruginosa*, *A. baumannii*	cUTI, pyelonephritis
Eravacycline	ESBL, CR *Enterobacteriaceae* spp., MRSA, *A. baumannii*, VRE	cUTI, cIAI
Meropenem-vaborbactam	ESBL, CR *Enterobacteriaceae* (Ambler Class A and C)	cUTI, pyelonephritis, cIAI, HAP, VAP, and BSI

ESBL—extended-spectrum βlactamase, CR—carbapenem resistant, VRE—vancomycin resistant Enterobacteriaceae, cUTI—complicated urinary tract infection, cIAI—complicated intrabdominal infections, HAP—hospital acquired pneumonia, VAP—ventilator associated pneumonia, BSI–bloodstream infections.

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
