# Peer review of "Trends, Epidemiology, and Management of Multi-Drug Resistant Gram-Negative Bacterial Infections in the Hospitalized Setting"

_antibiotics, 2020, doi:10.3390/antibiotics9040196_

Round 1

Reviewer 1 Report

The abstract needs major revision, avoiding hyperbole and emotive language. The  in the remainder of the paper is well written.

The section on antibiotic stewardship is focussed on countries with well developed and regulated  medical systems. A huge problem in Asian countries is that antibiotics are available over the counter, without prescription, leading to inappropriate and incomplete use of antibiotics. Secondly the large scale use of antibiotics in agricultural settings, which leads to massive release into the environment. These issues are not mentioned sufficiently (mentioned only briefly in the section on Colistin) , yet they are very important.  There have been papers published about antibiotic release into the environment and these need to be cited more extensively by the authors in their section antibiotic stewardship. The failure of the consumer to complete the course of antibiotics is another critical issue to which adequate attention has not been given.

Do the authors have permission to reproduce Figure 2 from Rogers ??

The review of the various classes of antibiotics was good .

I have attach the original file with some suggested changes to the abstract. 

Author Response

Responses to Reviewer 1:

Thank you for your thoughtful and insightful comments.

  1. The abstract needs major revision, avoiding hyperbole and emotive language. The remainder of the paper is well written. We have adjusted the abstract and hope it better reflects the intent of the paper. We unfortunately did not see the attachment with your modifications and would be open to accepting that version in its stead as well. “Antibiotic resistance is a serious threat to the health of humanity, particularly vulnerable populations and among gram-negative bacteria. This leads to increasing healthcare costs, morbidity and mortality.   Ever-evolving, these bacteria are developing new and improved mechanisms of resistance and methods of antimicrobial evasion at a rapid rate. Escherichia coli, Pseudomonas aeruginosa, Klebsiella pneumoniae, and Acinetobacter baumannii have all been identified as pathogens with particularly high rates of resistance in the face of a dwindling pool of available treatments. Effectively combating this issue requires both preventative and reactive measures. Reducing the spread of resistant pathogens as well as the evolution of resistance is complicated. Such a task requires a more judicious use of antibiotics through a better understanding of infection epidemiology, resistance patterns, and guidelines for treatment. These goals can best be achieved through the implementation of antimicrobial stewardship programs and the development and institution of new drugs capable of eradicating MDR gram-negative pathogens. The purpose of this article is to review current trends in MDR gram-negative bacterial infections as well as current guidelines for management. Finally, it will discuss new and emerging antimicrobials as well as future considerations for antibiotic resistance on a global scale.”
  2. The section on antibiotic stewardship is focused on countries with well-developed and regulated  medical systems. A huge problem in Asian countries is that antibiotics are available over the counter, without prescription, leading to inappropriate and incomplete use of antibiotics. We agree that these important issues should be discussed in more detail and have included a section under antibiotic stewardship.
  3. Secondly the large scale use of antibiotics in agricultural settings, which leads to massive release into the environment. These issues are not mentioned sufficiently (mentioned only briefly in the section on Colistin) , yet they are very important.  There have been papers published about antibiotic release into the environment and these need to be cited more extensively by the authors in their section antibiotic stewardship. The failure of the consumer to complete the course of antibiotics is another critical issue to which adequate attention has not been given. Thank you for pointing this out. We included an expanded section on stewardship to address the reviewer's concerns.
  4. Do the authors have permission to reproduce Figure 2 from Rogers? Yes, this illustration was created expressly from this institution. Karlee Rogers is at Rowan University as well. Please advise if you require any specific documentation to this effect.
  5. The review of the various classes of antibiotics was good. Thank you!
  6. I have attached the original file with some suggested changes to the abstract. Our apologies but we did not see the attached document with changes.

Reviewer 2 Report

The title is broad and it sounds very promising. However, the manuscript does not review the latest trends thoroughly enough, and focuses only on hospital-acquired infections and pathogens. Furthermore, the manuscript is neither polished nor proofread - there are a lot of mistakes, unclear sentences, and inconsistencies. As infections by gram-negative bacteria are extremely important, I strongly encourage the authors to address some major issues (major and some minor remarks listed below), re-write, and re-submit the manuscript. For now, I recommend either rejection or major revision with extended deadline, so the authors have enough time to fix the manuscript.

Major remarks:

1. Two very important topics are not covered at all: phage therapy and vaccines. Even though phage therapy is not a new concept, it is re-gaining importance in the "post-antibiotic era", and it has been successfully applied  a few years ago (again, after decades) against severe infections with P. aeruginosa, resistant to all available antibiotics.

2. Sixty three references may be sufficient for a research article, but definitely not for a serious review.

3. The manuscript is narrower than promised in L18-21 and in the title. According to the introduction, abstract and title, the focus is on gram-negative, and yet we learn nothing about MDR H. pylori, Campylobacter, Neisseria and other very important gram-negative bacteria. In fact, we learn more about certain gram-positive bacteria than gram-negatives. Later in the manuscript, the authors explicitly mention that the focus of a specific section is on HAI. However, I would appreciate if this was declared earlier. 

4. The flow/structure of this manuscript is neither clear nor intuitive. Throughout the first part, I was struggling with the purpose of this review. It was not completely clear if the main focus was HAI or gram-negative infections in general. Additionally, it was not clear to me if the authors were communicating medical textbook information with brief updates, and whether the authors were talking about USA alone or the whole world (e.g. L42-43 and other cost-related sections). It would definitely be interesting to see how the infections increase costs in other countries with better/worse health insurance, and if there is any research done on the influence of "big pharma". I am aware that the last subject is very political and kind of sensitive, but we saw in the news what happened to the prices of certain medicines for HIV+ patients.

5. The chapter Mechanisms of resistance is very textbook-like, and does not provide enough recent findings. What is new in this field? Besides, this chapter covers only beta-lactamases and LPS changes, and it does not go into depth, novelties, and latest trends. Other mechanisms are not described, except depicted in Figure 2, and they are of extreme importance. The purpose of Figure 1 (which should in fact be Table 1) is also unclear to me, as the table lacks very important information: Where are these enzymes present? References. Are these enzymes (I guess) from different organisms phylogenetically related or homologous? 

6. Risk factors: Was recently any research done on how disinfection/sterilization techniques affect resistance in G- bacteria, like in Staphylococcus (e.g. https://www.frontiersin.org/articles/10.3389/fmicb.2018.02664/full)?  Thorougher research review on risk factors for resistance would be greatly appreciated. 

7. Treatment: It is unclear if guidelines are the same in all countries, on all continents. If not, what are the differences? What is new compared to the old guidelines? BSI: what are the guidelines in treatment? The whole treatment section could be shortened, and updated with new facts, without textbook classifications and some very basic descriptions. Therapeutic approaches: any new research on synergy? As basic research and clinical studies on novel antimicrobial drugs are very expensive and longitudinal (it may take more than a decade for a compound to be commercially available), I strongly believe that the major focus in this section should be on "other therapeutic approaches". Antibiotic resistance is a major problem, and some bacteria adjust to novel compounds extremely quickly (e.g. Neisseria). Even if we get approved anti-Neisseria drugs now, in about 20 years, a lot of the isolates will be resistant to them. And we will be in the same situation.

9. I recommend the authors to carefully proofread and polish the manuscript. There are several sentences that do not make sense (e.g. L6-7: "Antibiotic resistance is a serious threat to the health of humanity, particularly vulnerable populations and among gram-negative bacteria."; L40-41: "Concerning bacteria are the lowest priority on the list and were entirely gram-positive bacteria in 2019."), certain species names are not written according to the standard nomenclature (e.g. Figure 3 and throughout the text), tables are labeled as figures (Figs. 1, 3, 4), figures are cited as Figure x, articles are not used correctly in some cases (e.g. L25: "Antibiotic resistance is growing problem around the world..."), some references are missing. It seems like certain facts are not cited at all or are cited inappropriately and not enough (e.g. first paragraph of the introduction). 

Some minor remarks:

L6-7: The first sentence of the abstract does not make sense. Please re-phrase it.

L16: These are solutions only for reactive measures.

L18: Will new drugs really solve the problem of MDR bacteria? Maybe on a short run. But very soon, we will be in the same situation.

L37 and throughout: Use the greek letter beta.

Throughout: make sure that the names of genera and species are written according to the standard bacterial nomenclature.

L63: Define UTI.

L65: Define ICU.

L80: I cannot find the definition of GNB in the text before this line.

L87: Sub-inhibtory is more appropriate term.

L91: Figure X?

L112: Tautology. If they are ubiquitous, they are everywhere. In the environment and "non-environment".

L115-119: Similar to N. gonorrhoeae and some other bacteria. Is this mechanism still unique? 

L121-123: These two articles were published in 2016 and 2017. What is new in the exciting field of adjunctive therapy and synergy studies? 

L127: Group is more appropriate than family, as these bacteria are not phylogenetically related.

L146: Contribute in singular or correct punctuation.

L178: Reference for this "ideal" missing. "positively identified" is an oxymoron. 

L187: 55- superscript? 

L249-267: This does not belong to the treatment but to risk factors. 

L298: Figure X???

Author Response

The title is broad and it sounds very promising. However, the manuscript does not review the latest trends thoroughly enough, and focuses only on hospital-acquired infections and pathogens. Furthermore, the manuscript is neither polished nor proofread - there are a lot of mistakes, unclear sentences, and inconsistencies. As infections by gram-negative bacteria are extremely important, I strongly encourage the authors to address some major issues (major and some minor remarks listed below), re-write, and re-submit the manuscript. For now, I recommend either rejection or major revision with extended deadline, so the authors have enough time to fix the manuscript.

Major remarks:

  1. Two very important topics are not covered at all: phage therapy and vaccines. Even though phage therapy is not a new concept, it is re-gaining importance in the "post-antibiotic era", and it has been successfully applied  a few years ago (again, after decades) against severe infections with P. aeruginosa, resistant to all available antibiotics.
    1. Thank you for this insight. We added an additional section on Nanosecond Electrical Pulses (EPs), phage therapy, Visible Light Therapy (VLT) and Photothermal Therapy under therapeutic approaches
  2. Sixty three references may be sufficient for a research article, but definitely not for a serious review.
    1. Additional references were added. There are now 79 references.
  1. The manuscript is narrower than promised in L18-21 and in the title. According to the introduction, abstract and title, the focus is on gram-negative, and yet we learn nothing about MDR H. pylori, Campylobacter, Neisseria and other very important gram-negative bacteria. In fact, we learn more about certain gram-positive bacteria than gram-negatives. Later in the manuscript, the authors explicitly mention that the focus of a specific section is on HAI. However, I would appreciate if this was declared earlier.
  2. The abstract has been revised to narrow the scope of this paper to multi-drug resistance gram negative bacterial infections in the hospitalized setting. This was the original intent of the paper. We apologize for vagaries.
  3. The flow/structure of this manuscript is neither clear nor intuitive. Throughout the first part, I was struggling with the purpose of this review. It was not completely clear if the main focus was HAI or gram-negative infections in general. Additionally, it was not clear to me if the authors were communicating medical textbook information with brief updates, and whether the authors were talking about USA alone or the whole world (e.g. L42-43 and other cost-related sections). It would definitely be interesting to see how the infections increase costs in other countries with better/worse health insurance, and if there is any research done on the influence of "big pharma". I am aware that the last subject is very political and kind of sensitive, but we saw in the news what happened to the prices of certain medicines for HIV+ patients.
  4. We have revised the abstract and verbiage throughout the article to narrow the scope toward MDR GNB infections in the hospitalized and acute care setting. Data reported on the economic burden has been US based. When studies or data reported are based in other countries, special note is made.
  5. We agree with the reviewers concerns about pharmaceutical influence but felt that discussing insurance and pharmaceutical issues would require expansion of the paper.
  6. The chapter Mechanisms of resistance is very textbook-like, and does not provide enough recent findings. What is new in this field? Besides, this chapter covers only beta-lactamases and LPS changes, and it does not go into depth, novelties, and latest trends. Other mechanisms are not described, except depicted in Figure 2, and they are of extreme importance. The purpose of Figure 1 (which should in fact be Table 1) is also unclear to me, as the table lacks very important information: Where are these enzymes present? References. Are these enzymes (I guess) from different organisms phylogenetically related or homologous? 
  7. a. We appreciate the reviewer’s comments and have modified the text.
  8. Risk factors: Was recently any research done on how disinfection/sterilization techniques affect resistance in G- bacteria, like in Staphylococcus (e.g. https://www.frontiersin.org/articles/10.3389/fmicb.2018.02664/full)?  Thorougher research review on risk factors for resistance would be greatly appreciated. 
  9. We added sections on Nanosecond Electrical Pulses (EPs), phage therapy, Visible Light Therapy (VLT) and Photothermal Therapy 
  10. Treatment: It is unclear if guidelines are the same in all countries, on all continents. If not, what are the differences? What is new compared to the old guidelines? BSI: what are the guidelines in treatment? The whole treatment section could be shortened, and updated with new facts, without textbook classifications and some very basic descriptions. Therapeutic approaches: any new research on synergy? As basic research and clinical studies on novel antimicrobial drugs are very expensive and longitudinal (it may take more than a decade for a compound to be commercially available), I strongly believe that the major focus in this section should be on "other therapeutic approaches". Antibiotic resistance is a major problem, and some bacteria adjust to novel compounds extremely quickly (e.g. Neisseria). Even if we get approved anti-Neisseria drugs now, in about 20 years, a lot of the isolates will be resistant to them. And we will be in the same situation.
  11. The purpose of this review was to discuss current guidelines and recommendations for the treatment of MDR GNB infections. These recommendations were based largely on IDSA and US CDC guidelines. We will include a section on other therapeutic approaches as future considerations and we will expand on synergy in those sections.
  12. I recommend the authors to carefully proofread and polish the manuscript. There are several sentences that do not make sense (e.g. L6-7: "Antibiotic resistance is a serious threat to the health of humanity, particularly vulnerable populations and among gram-negative bacteria."; L40-41: "Concerning bacteria are the lowest priority on the list and were entirely gram-positive bacteria in 2019."), certain species names are not written according to the standard nomenclature (e.g. Figure 3 and throughout the text), tables are labeled as figures (Figs. 1, 3, 4), figures are cited as Figure x, articles are not used correctly in some cases (e.g. L25: "Antibiotic resistance is growing problem around the world..."), some references are missing. It seems like certain facts are not cited at all or are cited inappropriately and not enough (e.g. first paragraph of the introduction). We have addressed changes in the text.

Some minor remarks:

L6-7: The first sentence of the abstract does not make sense. Please re-phrase it. Sentence has been rephrased.

L16: These are solutions only for reactive measures. ?

L18: Will new drugs really solve the problem of MDR bacteria? Maybe on a short run. But very soon, we will be in the same situation.

L37 and throughout: Use the greek letter beta. Adjustments have been made.

Throughout: make sure that the names of genera and species are written according to the standard bacterial nomenclature.

L63: Define UTI. This has been defined.

L65: Define ICU. This has been defined.

L80: I cannot find the definition of GNB in the text before this line. This has been defined in the introduction.

L87: Sub-inhibtory is more appropriate term. This revision has been made.

L91: Figure X? This has been revised.

L112: Tautology. If they are ubiquitous, they are everywhere. In the environment and "non-environment". ?

L115-119: Similar to N. gonorrhoeae and some other bacteria. Is this mechanism still unique? 

L121-123: These two articles were published in 2016 and 2017. What is new in the exciting field of adjunctive therapy and synergy studies? 

L127: Group is more appropriate than family, as these bacteria are not phylogenetically related.

L146: Contribute in singular or correct punctuation.

L178: Reference for this "ideal" missing. "positively identified" is an oxymoron. 

L187: 55- superscript? 

L249-267: This does not belong to the treatment but to risk factors. 

L298: Figure X??? This has been revised.

Round 2

Reviewer 1 Report

The abstract still needs improvement. I sent a revised abstract and the editors should have forwarded this to the authors.

Here is a revised version without hyperbole.:

 Abstract: The increasingly prevalence of antibiotic resistance in bacteria is a threat to human health, particularly within  vulnerable populations in  hospital and acute care settings. This leads to increasing healthcare costs, morbidity and mortality. Bacteria rapidly evolve novel mechanisms of resistance and methods of antimicrobial evasion. Escherichia coli, Pseudomonas aeruginosa, Klebsiella pneumoniae and Acinetobacter baumannii have all been identified as pathogens with particularly high rates of resistance to antibiotics, resulting in a reducing pool of available treatments for these organisms. Effectively combating this issue requires both preventative and reactive measures. Reducing the spread of resistant pathogens, as well as reducing the rate of evolution of resistance are complex tasks. Such tasks require a more judicious use of antibiotics through a better understanding of epidemiology, resistance patterns and better implementation of guidelines for treatment. These goals can best be achieved  through the implementation of antimicrobial stewardship programs and the development and introduction of new drugs capable of eradicating multi-drug resistant gram-negative pathogens (MDR GNB). The purpose of this article is to review recent trends in MDR gram-negative bacterial infections in the hospital setting, as well as current guidelines for management. Finally,  new and emerging antimicrobials as well as future considerations for combating antibiotic resistance on a global scale are discussed.

Author Response

Here is a revised version without hyperbole.:

 Abstract: The increasingly prevalence of antibiotic resistance in bacteria is a threat to human health, particularly within vulnerable populations in hospital and acute care settings. This leads to increasing healthcare costs, morbidity and mortality. Bacteria rapidly evolve novel mechanisms of resistance and methods of antimicrobial evasion. Escherichia coli, Pseudomonas aeruginosa, Klebsiella pneumoniae and Acinetobacter baumannii have all been identified as pathogens with particularly high rates of resistance to antibiotics, resulting in a reducing pool of available treatments for these organisms. Effectively combating this issue requires both preventative and reactive measures. Reducing the spread of resistant pathogens, as well as reducing the rate of evolution of resistance are complex tasks. Such tasks require a more judicious use of antibiotics through a better understanding of epidemiology, resistance patterns and better implementation of guidelines for treatment. These goals can best be achieved through the implementation of antimicrobial stewardship programs and the development and introduction of new drugs capable of eradicating multi-drug resistant gram-negative pathogens (MDR GNB). The purpose of this article is to review recent trends in MDR gram-negative bacterial infections in the hospital setting, as well as current guidelines for management. Finally, new and emerging antimicrobials as well as future considerations for combating antibiotic resistance on a global scale are discussed.

Thank you for your revision. This has been amended.

Reviewer 2 Report

The latest version of the manuscript still needs some language polishing and proofreading. After the authors address my comments, I can endorse the manuscript for publication.

Major remarks:

1. Title: please specify that the article is talking about hospital infections.

2. Table 1: I still find this table problematic, and this comment was not addressed in authors' response. Please add the names of the organisms that encode these enzymes. These enzyme names are not informative at all.

3. Figures and tables. Currently, the manuscript contains Table 1, Figure 2-4. Figures 3 and 4 should be captioned as tables. Figure 2 should be Figure 1.

Minor remarks:

L37, 60, 61, 407, 506, 541 AND THROUGHOUT: beta (symbol), not b! please unify!

L74: fosfomycin not capitalized

L87-88: Bacteria evolve all the time, not only in these circumstances. Additionally, I would delete "inappropriate" 

Table 1: I still find this table problematic, and this comment was not addressed in authors' response. Please add the names of the organisms that encode these enzymes. These enzyme names are not informative at all.

Table 1 caption: this table is not a description but a list or classification or something else. it does not describe anything.

L124: please replace "bug" with bacterium

L180: I am not a physician, but I would also tailor it to the patient (allergies, kidney/liver failure etc)

L193: please spell out the abbreviations in titles

Figure 3=table. Enterococcus spp. (spp. missing)

Figure 4=table. It is NOT a description. Please re-think all captions.

L438, 460, 477 and throughout: please italicize all bacterial names and write them according to standards (species name is NOT capitalized like e.g. in L460).

L440 and throughout: they were synergistic or in synergy. They did not produce synergy.

L473-5: reference for "recent study" missing

L565 and throughout: please unify the volumetric units. make a decision about capitalizing (or not) and stick to it.

L587: Capitalize the first letter in the sentence

L599: proteins or peptides? They are not interchangeable.

L655 and throughout: if ns stands for nanosecond, there should be non-breaking space between the value and the unit.

L657: please use the greek letter mu/my for micro.

L660 and throughout: italicize in vitro/in vivo

L712: space

Author Response

The latest version of the manuscript still needs some language polishing and proofreading. After the authors address my comments, I can endorse the manuscript for publication.

Major remarks:

  1. Title: please specify that the article is talking about hospital infections.

This has been revised. Thank you.

  1. Table 1: I still find this table problematic, and this comment was not addressed in authors' response. Please add the names of the organisms that encode these enzymes. These enzyme names are not informative at all. Gram negative organisms which produce these enzymes have been added to the table. Thank you.

  1. Figures and tables. Currently, the manuscript contains Table 1, Figure 2-4. Figures 3 and 4 should be captioned as tables. Figure 2 should be Figure 1. This has been revised. Thank you.
  2.  

Minor remarks:

L37, 60, 61, 407, 506, 541 AND THROUGHOUT: beta (symbol), not b! please unify! This has been revised. Thank you.

L74: fosfomycin not capitalized Revised. Thank you.

L87-88: Bacteria evolve all the time, not only in these circumstances. Additionally, I would delete "inappropriate" This sentence has been clarified and “inappropriate” has been removed. Thank you.

Table 1: I still find this table problematic, and this comment was not addressed in authors' response. Please add the names of the organisms that encode these enzymes. These enzyme names are not informative at all. Gram negative organisms which produce these enzymes have been added to the table. Thank you.

Table 1 caption: this table is not a description but a list or classification or something else. it does not describe anything. The word description has been removed. Thank you.

L124: please replace "bug" with bacterium This has been revised. Thank you.

L180: I am not a physician, but I would also tailor it to the patient (allergies, kidney/liver failure etc) This has been added. Thank you for the suggestion.

L193: please spell out the abbreviations in titles This has been revised. Thank you.

Figure 3=table. Enterococcus spp. (spp. missing) This has been corrected. Thank you.

Figure 4=table. It is NOT a description. Please re-think all captions. This has been addressed. Thank you.

L438, 460, 477 and throughout: please italicize all bacterial names and write them according to standards (species name is NOT capitalized like e.g. in L460). This has been addressed. Thank you.

L440 and throughout: they were synergistic or in synergy. They did not produce synergy. This has been addressed. Thank you.

L473-5: reference for "recent study" missing This was from reference 40. The in-text citation was added. Thank you.  

L565 and throughout: please unify the volumetric units. make a decision about capitalizing (or not) and stick to it. This has been addressed. Thank you.

L587: Capitalize the first letter in the sentence. This has been addressed. Thank you.

L599: proteins or peptides? They are not interchangeable. This has been addressed. Thank you.

L655 and throughout: if ns stands for nanosecond, there should be non-breaking space between the value and the unit. This has been addressed. Thank you.

L657: please use the greek letter mu/my for micro. This has been addressed. Thank you.

L660 and throughout: italicize in vitro/in vivo This has been addressed. Thank you.

L712: space This has been addressed. Thank you.